# SBFA: Single Sneaky Bit Flip Attack to Break Large Language Models

## Abstract

Model integrity of Large language models (LLMs) has become a pressing security concern with their massive online deployment. Prior Bit-Flip Attacks (BFAs)—a class of popular AI weight memory fault-injection techniques—can severely compromise Deep Neural Networks (DNNs): as few as tens of bit flips can degrade accuracy toward random guessing. Recent studies extend BFAs to LLMs and reveal that, despite the intuition of better robustness from modularity and redundancy, only a handful of adversarial bit flips can also cause LLMs' catastrophic accuracy degradation. However, existing BFA methods typically focus on either integer or floating-point models separately, limiting attack flexibility. Moreover, in floating-point models, random bit flips often cause perturbed parameters to extreme values (e.g., flipping in exponent bit), making it not stealthy and leading to numerical runtime error (e.g., invalid tensor values (NaN/Inf)). In this work, for the first time, we propose SBFA (Sneaky Bit-Flip Attack), which collapses LLM performance with only one single bit flip while keeping perturbed values within benign layer-wise weight distribution. It is achieved through iterative searching and ranking through our defined parameter sensitivity metric, *ImpactScore*, which combines gradient sensitivity and perturbation range constrained by the benign layer-wise weight distribution. A novel lightweight SKIP searching algorithm is also proposed to greatly reduce searching complexity, which leads to successful SBFA searching taking only tens of minutes for SOTA LLMs. Across Qwen, LLaMA, and Gemma models, with only **one single bit flip**, SBFA successfully degrades accuracy to below random levels on MMLU and SST-2 in both BF16 and INT8 data formats. Remarkably, flipping a single bit out of billions of parameters reveals a severe security concern of SOTA LLM models.

## 1 Introduction

Large Language Models (LLMs) have emerged as a transformative paradigm in natural language processing, demonstrating remarkable capabilities across a wide spectrum of tasks such as text understanding, generation, translation, and reasoning. Their success is largely attributed to advances in large-scale pretraining, transformer architectures, and scaling laws, which enable them to capture rich linguistic and semantic patterns from massive text corpora, making them powerful general-purpose tools for language technologies (Minaee et al., 2024). However, the increasing deployment of LLMs in critical applications has raised significant security concerns, particularly in adversarial settings where malicious actors may seek to exploit vulnerabilities in these model weight parameters to compromise their integrity, confidentiality, or availability (Das et al., 2025).

One prominent vulnerability of LLMs is their susceptibility to Bit-Flip Attacks (BFAs)—a class of weight memory fault injection techniques that could only flip tens of binary bits out of millions of AI model weight parameters stored in computer main memory to induce erroneous behavior (Yao et al., 2020; Rakin et al., 2019; 2021). While BFAs have been extensively studied and demonstrated in the context of Deep neural networks (DNNs) (Yao et al., 2020; Rakin et al., 2020), recent works have extended these attacks to LLMs (Das et al., 2024; Almalky et al., 2025). Despite earlier assumptions that the massive scale, modularity, and redundancy of LLMs would provide inherent robustness, emerging evidence shows that LLMs remain highly vulnerable. These models often comprise over 100× more parameters than DNNs(e.g., LLaMA3.1-8B-Instruct with 8 billion parameters (Meta AI,

2024) vs. ResNet-50 with 25 million (He et al., 2016)), yet even a few bit flips can cause catastrophic degradation in performance.

However, existing BFA approaches for LLMs primarily focus on either integer or floating-point models separately, which limits their applicability across diverse deployment settings. In floating-point format, bit flips can drive parameters to extreme values. For example, flipping an exponent bit may yield invalid tensor values (NaN/Inf) during computation that trigger numerical runtime errors—making it thus not stealthy.

In this work, we are the first to propose SBFA (Sneaky Bit-Flip Attack)—a novel bit-flip attack that can completely malfunction State-of-the-art (SOTA) LLMs with just one single bit flip, while remaining stealthy and effective across both floating-point and integer quantized models. To identify such one single most critical bit out of hundreds of billions LLM model parameters, SBFA introduces a new parameter sensitivity metric, called ImpactScore, which integrates weight gradient sensitivity with perturbation constrained by the benign layer-wise weight distribution to search for the most critical bit. The main technical contributions of this work are:

- For the first time, we introduce *SBFA*, a novel sneaky bit-flip attack that can compromise the performance of LLMs close to random guess with just a single bit flip, while restricting the perturbed parameter still remain within benign layer-wise weight distribution for stealthiness.

- We propose a new metric, called *ImpactScore*, to evaluate weight parameter vulnerability. It combines gradient sensitivity and perturbation range constrained by the benign layer-wise weight distribution. This score captures both the importance of specific parameter to the model's performance through first-order gradient and the feasibility of perturbing it within benign weight range.

- To improve searching efficiency, we propose *SKIP (Selective sKipping with Impact Prioritization) Search*, a lightweight searching algorithm that greatly accelerates the identification of critical bits to flip by selectively skipping low-impact parameters and layers. This significantly reduces the computational overhead of the attack while maintaining its effectiveness.

- We conduct extensive experiments across multiple LLM architectures, including Qwen, LLaMA, and Gemma, with both popular BF16 and INT8 data formats. The experiment results demonstrate that our SBFA can degrade LLM performance to near-random guess with only one single bit flip on benchmarks MMLU and SST-2. These impressive results underscore the alarming vulnerability of LLMs against adversarial weight fault injection.

## 2 BACKGROUND

### 2.1 BIT FLIP FAULT IN COMPUTER MEMORY THROUGH ROWHAMMER ATTACK

The well-known RowHammer attack demonstrated in real computer memory revealed that repeatedly activating a row in Dynamic Random Access Memory (DRAM) can induce disturbance bit flip errors in physically adjacent rows, effectively "flipping" bits without directly accessing them (Kim et al., 2014). This hardware-level fault, caused by charge leakage due to frequent row activations, demonstrated that memory cells are more vulnerable than previously assumed and opened a new class of security exploits where attackers can manipulate system behavior at the memory bit level. This Rowhammer attack lays the hardware foundation of adversarial bit-flip attacks (Yao et al., 2020; Rakin et al., 2019; 2021).

### 2.2 BIT-FLIP ATTACK (BFA) AND RELATED WORKS

**Bit-Flip Attack (BFA) on DNNs.** Bit-Flip Attack (BFA) (Rakin et al., 2019) is a fault-injection technique that exploits hardware memory vulnerabilities (e.g., RowHammer) to alter the binary representation of neural network parameters, thereby inducing erroneous behavior in the model. Previously, BFA methods typically deployed gradient-based analysis to estimate the sensitivity of each weight bit, identifying those whose corruption most significantly increases the model loss. By strategically flipping as few as three of vulnerable bits, attackers can drastically degrade model

accuracy (Rakin et al., 2019; Yao et al., 2020). Also, later studies applied BFAs to DNNs to induce targeted (trojan) outputs for specific inputs (Rakin et al., 2020). The effectiveness of BFAs highlights how leveraging gradient information enables highly targeted and efficient parameter corruption, making them a powerful threat to the integrity and security of DNNs.

**Bit-Flip Attack (BFA) on LLMs.** However, BFAs on LLMs have received limited exploration. A recent study (Almalky et al., 2025) adapted the basic gradient-based BFA to LLMs but restricted the attack to the most significant bit (MSB) in floating-point weights. Their results suggested that LLMs exhibit relative tolerance to such attacks, with no obvious performance degradation. In contrast, GenBFA (Das et al., 2024) targeted INT8 quantized models and demonstrated that LLMs can be highly vulnerable, requiring only a few bit flips (e.g., 3–10) to cause significant accuracy drops.

**Limitations of Existing BFA Methods.** Nonetheless, both approaches are constrained to specific numerical formats (either FP32/FP16 or INT8), limiting their generalizability. Moreover, in floating-point settings, bit flips often drive parameters to extreme values (e.g., out of the benign weight distribution), making such perturbation not stealthy or causing software numerical runtime errors during inference or training. Therefore, there is a need for a more efficient and stealthy method to attack LLMs that can work across different numerical formats. This motivates our development of SBFA, a novel bit-flip attack that can compromise the integrity of LLMs with just a single bit flip, while remaining stealthy and effective across both floating-point and integer quantized models.

## 3 THREAT MODEL

Similar as previous BFA works, We consider a white-box threat model, where the attacker has full access to the victim LLM, including its architecture and parameters. The attacker can compute gradients with respect to a small set of input samples to identify vulnerable bits for flipping (Kim et al., 2014; Rakin et al., 2019). The attacker's goal is to degrade the model's performance on the target downstream tasks by flipping as few bits as possible, ideally just one single bit, while ensuring the perturbed weights remain within benign weight distribution for stealthiness.

## 4 SNEAKY BIT-FLIP ATTACK (SBFA) METHOD

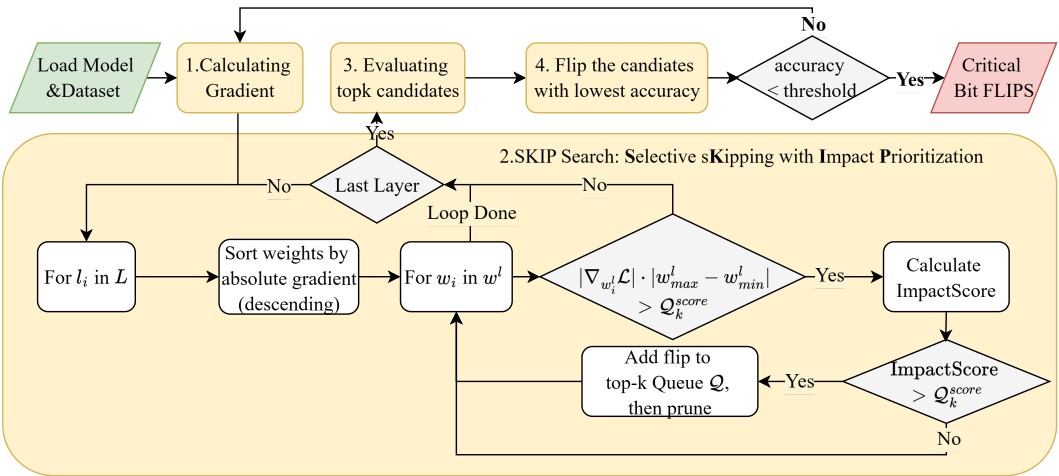

Figure 1: Proposed SBFA searching workflow.

In this section, we introduce the proposed SBFA searching algorithm to identify critical bit flips for target LLM, illustrated in Figure 1. The procedure begins by loading the target LLM and dataset, followed by computing the gradient of each weight. Next, all weight parameters and their potential sneaky bit flips are ranked using our defined ImpactScore metric to generate a global top-$k$ candidate queue $\mathcal{Q}$. To improve searching efficiency, we propose SKIP search algorithm that could greatly reduce searching cost that will be described in the section 4.3. The resulting top-$k$ candidates are

then evaluated for their effects on degrading model accuracy using the test dataset, and the candidate that yields the largest degradation is the final most critical bit flip. This process iterates until the model's accuracy falls below a predefined threshold.

## 4.1 SNEAKY BIT-FLIP RANGE

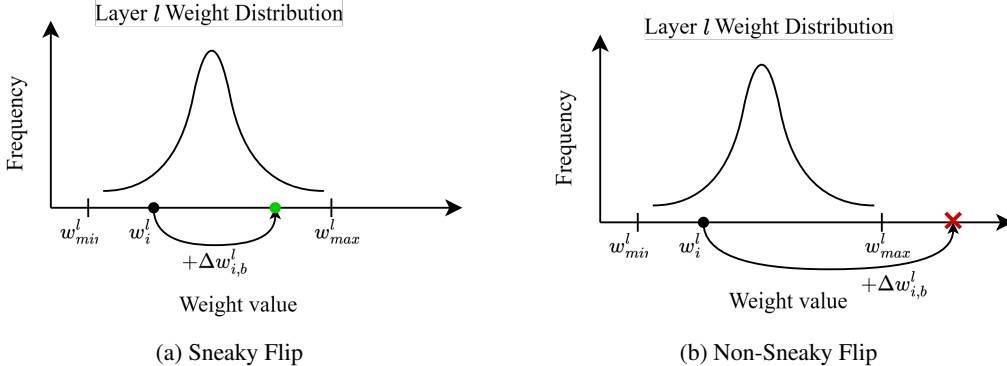

(a) Sneaky Flip          (b) Non-Sneaky Flip

Figure 2: Illustration of Sneaky Range Constraint. A flip is sneaky if the perturbed weight remains within the benign range $[w_{\min}^l, w_{\max}^l]$ (left), and non-sneaky otherwise(right).

Traditional BFAs may push the perturbed weights (i.e., after bit flip) to extremely large positive value or small negative value. It is common especially in floating-point formats where flipping certain exponent bit can cause numerical runtime errors. Such perturbations are not stealthy and could be easily detected through analyzing the layer-wise weight distribution. Therefore, in this work, to make our attack work for both integer and floating-point data formats, as shown in Fig.2, we define the **Sneaky Bit-Flip Attack (SBFA)** to enforce the bit-flipped (i.e., perturbed) weight parameter value remain within the layer weight distribution. It means the perturbed weight parameter will not be larger (or smaller) than the benign maximum (or minimum) weight value within the same layer, to guarantee the stealthiness of the injected fault. Formally, for a weight $w_i^l$ in layer $l$, a candidate flip at bit position $b$ induces change $\Delta w_{i,b}^l$. The flip is sneaky only if

$$w_{\min}^l \leq w_i^l + \Delta w_{i,b}^l \leq w_{\max}^l \tag{1}$$

where $w_{\min}^l$ and $w_{\max}^l$ are the minimum and maximum weight values in that layer.

## 4.2 IMPACTSCORE

To efficiently identify the most critical bit to flip, we introduce a novel metric, called **ImpactScore**, that quantifies the potential impact of flipping the selected bit in the model parameters. The ImpactScore combines two key factors: weight sensitivity captured by the gradient w.r.t. loss function and the largest possible magnitude of change induced by sneaky bit flip. It is defined as:

$$\text{ImpactScore}_i^l = \left| \nabla_{w_i^l} \mathcal{L} \right| \cdot \max_{b \in \mathcal{B}} \left( |\Delta w_{i,b}^l| \right), \quad \text{s.t.} \ \ w_{\min}^l \leq w_i^l + \Delta w_{i,b}^l \leq w_{\max}^l \tag{2}$$

where $\nabla_{w_i^l} \mathcal{L}$ is the gradient of the loss with respect to $w_i^l$, and $\Delta w_{i,b}^l$, $w_{\min}^l$, and $w_{\max}^l$ are defined in Eq. 1. Here, $b \in \mathcal{B}$ denotes a bit position in the binary representation of $w_i^l$, and $\mathcal{B}$ is the set of all valid bit positions under the given precision format (e.g., $|\mathcal{B}| = 16$ for FP16, $|\mathcal{B}| = 8$ for INT8). The ImpactScore is defined as the product of the absolute weight gradient and the maximum absolute weight change induced by flipping bit positions $b$, subject to the range constraint in Eq. 1. The gradient term captures how sensitive the loss is to changes in that weight, while the perturbation term reflects the largest feasible change that can be induced by a bit flip without violating the layer-wise weight distribution. By combining these two factors, the ImpactScore prioritizes bits whose flipping is likely to cause significant performance degradation while remaining stealthy.

To illustrate, consider a layer with range $[w_{\min}^l, w_{\max}^l] = [-1, 1]$ and an initial weight $w = 0.5$, encoded in FP16 as 0 01110 0000000000. Flipping the most significant exponent bit (01110 $\rightarrow$

11110) yields $w' = 32768 \notin [-1, 1]$, which causes a large but non-sneaky perturbation. Flipping the sign bit (0000000000 $\rightarrow$ 0000000001) gives $\tilde{w} = 0.500488 \in [-1, 1]$, a sneaky small change. Flipping the sign bit ($0 \rightarrow 1$) produces $\tilde{w} = -0.5 \in [-1, 1]$, which is sneaky and induces the largest in-range perturbation. This example shows how all sneaky bit-flip-induced changes are enumerated, their magnitudes $|\Delta w_{i,b}^l|$ computed, and the maximum magnitude multiplied by the corresponding gradient to obtain the ImpactScore.

## 4.3 SKIP SEARCH: SELECTIVE SKIPPING WITH IMPACT PRIORITIZATION

The proposed SKIP Search is summarized in Alg. 1 in Appendix A and illustrated in Fig. 1. We define $\mathcal{Q}$ as a global priority queue that maintains the top-$k$ bit-flip candidates with the highest ImpactScores. Note we set $k$ as 100 in all our experiments. $\mathcal{Q}_k^{score}$ denotes the score of kth item in $\mathcal{Q}$ (i.e., minimum ImpactScore in $\mathcal{Q}$). $|\Delta w_{\max}^l - \Delta w_{\min}^l|$ is the weight range of layer $l$. From Eq. 1, we can conlude that $|\Delta w_{i,b}^l| \leq |\Delta w_{\max}^l - \Delta w_{\min}^l|$. In other words, the perturbation magnitude is upper bound by the layer weight range for any weight in that layer. Due to the vast number of parameters in LLMs, computing the ImpactScore for each parameter is computationally infeasible. To address this issue, we introduce **SKIP Search**, which reduces complexity by selectively skipping low-impact parameters' ImpactScore calculation during bit candidate ranking.

As shown in Fig. 1, after obtaining weight gradients at the current epoch, the SKIP searching algorithm iterates over all layers sequentially. For a given layer $l$, all weights are sorted in descending order by their absolute gradients. For weight $w_i^l$, if the global top-$k$ queue $\mathcal{Q}$ is already full and the product of its absolute gradient with the layer's weight range is smaller than the minimum ImpactScore in $\mathcal{Q}$, it skips all remaining weights in that layer, since none remaining weights in this layer can produce a higher ImpactScore to replace the last candidate in $\mathcal{Q}$. If it is larger, the corresponding weight ImpactScore will be calculated and compared with the $\mathcal{Q}_k^{score}$. The corresponding bit candidate will be added to the global $\mathcal{Q}$ when a larger ImpactScore is found. Otherwise, the algorithm proceeds to the next weight. Note that, $\mathcal{Q}$ will be pruned after every insertion to retain only the top-$k$ candidates. As a result, SKIP Search efficiently skips the majority of weights while still identifying the top-$k$ candidates with the highest ImpactScores. For example, with SKIP Search, the number of ImpactScore evaluations on Qwen3-14B is reduced from 14.8 billion to only 15,569 —an over $9.5 \times 10^5$-fold reduction— enabling fast attack on large-scale LLMs.

## 5 EXPERIMENTS

### 5.1 EXPERIMENTAL SETUP

**Models and datasets.** We evaluate SBFA on a diverse suite of LLMs, including Qwen2.5-7B (Qwen Team, 2024), the Qwen3 series (1.7B, 4B, 8B, 14B) (Yang et al., 2025), LLaMA3.1-8B-Instruct (Meta AI, 2024), and Gemma3-12B (Team et al., 2025). These models span a wide range of architectures and sizes (1.7B to 14B parameters), enabling a comprehensive evaluation of SBFA's effectiveness. All models are accessed via the Hugging Face model hub (Wolf et al., 2020). We evaluate models in both BF16 and INT8-quantized formats. For INT8 quantization, we use the `BitsAndBytes` (BNB) library (Dettmers et al., 2022) to convert FP models into INT8-compatible formats. We default to BF16 for most experiments, as it offers improved numerical stability and broader dynamic range for certain models. We evaluate SBFA on multiple benchmarks, including MMLU (Hendrycks et al., 2021) and SST-2 (Socher et al., 2013). MMLU is a challenging multi-task benchmark spanning 57 diverse subjects, while SST-2 is a binary sentiment classification task. To assess generalization, we further test SBFA on GSM8K (Cobbe et al., 2021) and ARC-Easy (Clark et al., 2018) using the `lm-evaluation-harness` framework (Gao et al., 2024).

**Evaluation Metric and Attack Settings.** We use **accuracy (Acc)** as the primary evaluation metric across all benchmarks, measuring the percentage of correctly answered samples. For all tasks, Acc is computed based on exact match between the model output and the reference answer. To evaluate attack effectiveness, we define a **critical threshold** as the chance-level accuracy for each task. An attack is considered *successful* if the post-attack accuracy falls below this threshold. Specifically, the critical threshold is set to 25% for MMLU (4-way multiple choice) and 50% for SST-2 (binary classification). All results are reported from the best outcome over three runs with different prede-

fined random seeds. For each run, we use 200 examples to compute gradients and 100 examples for evaluation. The batch size is set to 1 due GPU memory constraint. The attack is terminated either when the accuracy drops below the critical threshold or after a maximum of 500 iterations. We set $k = 100$ for SKIP Search, balancing efficiency and effectiveness.

## 5.2 RESULTS AND ANALYSIS

Table 1: Results on MMLU and SST-2 across models under different precision settings. **Pre-ACC** refers to accuracy before attack; **Post-ACC** is the lowest accuracy after attack. **Mode** indicates the attack strategy: INT8 attacks only INT8 weights; MIXED attacks both INT8 and BF16 weights.

| Model/Dataset | Precision | | MMLU | | | | SST-2 | | | |
|---|---|---|---|---|---|---|---|---|---|---|
| | Type | Mode | Pre-ACC | Post-ACC | # Flip | # Crit-1Flip | Pre-ACC | Post-ACC | # Flip | # Crit-1Flip |
| Qwen2.5-7B | BF16 | – | 0.71 | 0.00 | 1 | 8 | 0.94 | 0.00 | 1 | 8 |
| | INT8 | INT8 | 0.63 | 0.00 | 1 | 2 | 0.93 | 0.00 | 1 | 1 |
| | | MIXED | 0.73 | 0.00 | 1 | 8 | 0.93 | 0.00 | 1 | 5 |
| Qwen3-1.7B | BF16 | – | 0.53 | 0.00 | 1 | 53 | 0.83 | 0.00 | 1 | 62 |
| | INT8 | INT8 | 0.48 | 0.00 | 1 | 2 | 0.83 | 0.00 | 1 | 31 |
| | | MIXED | 0.48 | 0.00 | 1 | 62 | 0.83 | 0.00 | 1 | 31 |
| Qwen3-4B | BF16 | – | 0.69 | 0.00 | 1 | 15 | 0.92 | 0.00 | 1 | 15 |
| | INT8 | INT8 | 0.71 | 0.00 | 1 | 2 | 0.97 | 0.00 | 3 | F |
| | | MIXED | 0.71 | 0.00 | 1 | 3 | 0.97 | 0.15 | 1 | 15 |
| Qwen3-8B | BF16 | – | 0.70 | 0.00 | 1 | 38 | 0.95 | 0.00 | 1 | 30 |
| | INT8 | INT8 | 0.67 | 0.05 | 1 | 2 | 0.96 | 0.00 | 1 | 1 |
| | | MIXED | 0.70 | 0.00 | 1 | 4 | 0.96 | 0.00 | 1 | 4 |
| Qwen3-14B | BF16 | – | 0.77 | 0.00 | 1 | 40 | 0.96 | 0.00 | 1 | 37 |
| | INT8 | INT8 | 0.79 | 0.08 | 1 | 2 | 0.95 | 0.00 | 1 | 2 |
| | | MIXED | 0.79 | 0.04 | 1 | 2 | 0.95 | 0.00 | 1 | 6 |
| Gemma3-12B | BF16 | – | 0.72 | 0.00 | 1 | 22 | 0.96 | 0.00 | 1 | 7 |
| | INT8 | INT8 | 0.64 | 0.04 | 5 | F | 0.98 | 0.43 | 7 | F |
| | | MIXED | 0.73 | 0.00 | 1 | 22 | 0.98 | 0.00 | 1 | 6 |
| Llama3.1-8B-Instruct | BF16 | – | 0.68 | 0.00 | 1 | 52 | 0.89 | 0.00 | 1 | 59 |
| | INT8 | INT8 | 0.69 | 0.00 | 1 | 2 | 0.86 | 0.00 | 1 | 9 |
| | | MIXED | 0.69 | 0.00 | 1 | 4 | 0.86 | 0.00 | 1 | 13 |

Table 1 presents the overall performance of SBFA across multiple LLMs and precision settings on the MMLU and SST-2 benchmarks. SBFA consistently degrades model accuracy to near-zero levels across both BF16 and INT8 formats. Notably, each attack requires only a **single bit flip**, underscoring the efficiency and effectiveness of the proposed method.

The **Mode** column differentiates between attack strategies. In the INT8 mode, attacks are restricted to quantized weights only, while in the MIXED mode, both quantized and residual floating-point tensors (FP16/BF16) can be targeted. This distinction follows the implementation of the BitsAndBytes (BNB) library (Dettmers et al., 2022), where 1D tensors such as biases and layer norms remain in floating-point format and are not quantized due to their negligible memory footprint. Consequently, these tensors are excluded in INT8 mode but included in MIXED. We view the MIXED mode as a more realistic threat model, since it reflects all parameters actually present in a deployed LLM.

The **# Flip** column reports the number of bit flips required to reach the lowest observed accuracy, or the point at which the program terminated due to instability such as numerical runtime errors. This metric captures attack efficiency in terms of minimal perturbations needed for maximal degradation. Empirically, SBFA is highly effective: in our experiments SBFA succeeds in all BF16 cases and in every MIXED case, and it succeeds in the majority of INT8 cases (11/14). In BF16 and MIXED modes the attack typically achieves maximal degradation with a single bit flip, highlighting both its effectiveness and stealthiness. Under the stricter INT8 mode, which excludes floating-point tensors, one flip is often sufficient, although in some cases up to seven flips are required.

The **# Crit-1Flip** column indicates how many distinct single-bit flips can individually push accuracy below the task-specific critical threshold. This number reflects the model's susceptibility to bit-level perturbations. For example, Qwen3-4B in BF16 format has 15 distinct critical bits on MMLU, suggesting high vulnerability, while Qwen3-4B in INT8 format has only 3, implying better robustness under INT8 quantization. If the model can not be degraded below the threshold with any single bit

flip, we denote this with "F" (for "Failed"). Analysis across models reveals several notable findings. First, both small and large models can be effectively compromised with just a single bit flip, which is striking given the scale of LLMs. For example, Qwen3-14B contains over 14B parameters. This highlights the extreme sensitivity of modern LLMs to bit-wise fault perturbations. Second, larger model size does not necessarily correlate with a higher number of critical bits. For instance, Qwen3-4B exhibits 15 critical bits in BF16 on MMLU, whereas the larger Qwen3-14B has 40. This suggests that vulnerability is influenced not only by scale. Similarly, models of comparable scale such as Qwen3-8B, Qwen2.5-7B, and LLaMA3.1-8B-Instruct, exhibit notably different numbers of critical bits. This observation suggests that factors beyond model size, including architecture design and pretraining data, play significant roles in determining a model's susceptibility to bit-flip attacks.

## 5.3 COMPARISON WITH PRIOR BIT-FLIP ATTACKS

Table 2: Comparison of SBFA with prior methods on MMLU and SST-2 for Llama3.1-8B-Instruct and Phi-3-mini-128k-Instruct in INT8. [†]GenBFA results are reported from Das et al. (2024). [*]Basic gradient-based BFA results are based on our implementation of the method from Rakin et al. (2019), with and without range-aware constraints.

| Model | Method | MMLU | | | SST-2 | | |
|---|---|---|---|---|---|---|---|
| | | Post-ACC | #Flip | #Crit-1Flip | Post-ACC | #Flip | #Crit-1Flip |
| Llama3-8B-Instruct | GenBFA[†] | 0.00 | 3 | F | - | - | - |
| | BFA (No-Range)[*] | 0.00 | 3 | F | 0.88 | 2 | F |
| | BFA (In-Range)[*] | 0.07 | 7 | F | 0.00 | 31 | F |
| | **SBFA (Ours)** | **0.00** | **1** | **39** | **0.00** | **1** | **3** |
| Phi-3-mini-128k-Instruct | GenBFA[†] | 0.00 | 4 | F | - | - | - |
| | BFA (No-Range)[*] | 0.23 | 2 | F | 0.00 | 2 | F |
| | BFA (In-Range)[*] | 0.23 | 2 | F | 0.00 | 1 | 1 |
| | **SBFA (Ours)** | **0.00** | **1** | **7** | **0.00** | **1** | **7** |

Table 3: Comparison of SBFA with prior methods on SST2 for Qwen-1.8B in FP32. [‡]BFA results are reported from Almalky et al. (2025). [*]Basic gradient-based BFA results are based on our implementation of the method from Rakin et al. (2019), with and without range-aware constraints.

| Model | Method | SST2 | | |
|---|---|---|---|---|
| | | Post-ACC | #Flip | #Crit-1Flip |
| Qwen-1.8B | BFA (Sign-only)[‡] | 0.94 | 500 | F |
| | BFA (No-Range)[*] | 0.73 | 4 | F |
| | BFA (In-Range)[*] | 0.01 | 13 | F |
| | **SBFA (Ours)** | **0.00** | **1** | **74** |

The effectiveness of SBFA is further validated through comparisons with prior bit-flip attack methods. These include GenBFA (Das et al., 2024), a gradient-based BFA restricted to sign-bit flips as reported in Almalky et al. (2025), and a gradient-based BFA adapted to LLMs, which we implemented following Rakin et al. (2019) in two variants: (1) BFA (No-Range): without range constraints and (2)BFA (In-Range): with the same range constraints as defined in Eq. 1. Table 2 reports results on MMLU and SST-2 for LLaMA3.1-8B-Instruct and Phi-3-mini-128k-Instruct under the INT8-MIXED setting. Since GenBFA does not provide results on SST-2, the corresponding entries are marked with a dash (—). Table 3 presents additional results on SST-2 for Qwen-1.8B in FP32, comparing SBFA with the BFA method from Almalky et al. (2025) and our two implemented BFA variants. For these experiments, we report results across three runs with different random seeds and present the best-performing outcomes.

For the `INT8` results in Table 2, GenBFA requires multiple flips (3–4) to significantly degrade performance, whereas SBFA consistently drives accuracy to zero with just a single flip, underscoring its superior efficiency in identifying high-impact bits. In contrast, BFA (No-Range) fails to succeed with a single flip and either requires multiple flips or terminates with runtime errors, highlighting its limited effectiveness. BFA (In-Range), while avoiding runtime errors due to the range constraint,

still demands more flips to succeed, though it occasionally achieves success with a single flip. Furthermore, both GenBFA and BFA typically expose only one set of critical flips per model, whereas SBFA identifies substantially more (e.g., up to 39 for LLaMA3.1-8B-Instruct on MMLU), revealing a much broader vulnerability surface.

For the FP32 results from Table 3, the BFA method from Almalky et al. (2025) proves ineffective, despite applying 500 bit flips, it fails to reduce accuracy below 94%. Additionally, the BFA (No-Range) stopped at epoch 4 due to numerical runtime error during loss calculation, caused by parameters being perturbed near the limits of FP32. This highlights a key limitation of traditional BFA approaches in floating-point settings and motivates the need for more stable attack methods that avoid triggering numerical errors. Such instability is rarely observed in INT8 settings, where the narrower dynamic range naturally bounds parameter magnitudes. However, BFA (In-Range) can still successfully attack the model within 13 flips without triggering errors, which demonstrates the importance of sneaky(range constrain). On top of that, SBFA achieves a complete accuracy drop to 0% with just a single bit flip , demonstrating superior effectiveness in the floating-point settings. Furthermore, SBFA identifies 71 critical bits for Qwen-1.8B on SST-2, compared to zero discovered by the prior BFA method—revealing a much broader vulnerability surface under FP32.

## 5.4 RUNTIME AND SCALABILITY

Table 4: Runtime comparison of SBFA across different model sizes within the same family.

| Phase | Task | Time (seconds) | |
| --- | --- | --- | --- |
| | | Qwen3-1.7B | Qwen3-14B |
| 1 | Data/Model Load + Initial Setup | 50.16 | 76.21 |
| 2 | Calculating Gradient | 22.58 | 67.17 |
| 3 | Rank Top-$k$ Candidates | 75.14 | 137.91 |
| 4 | Evaluation of Candidate Flips | 559.37 | 787.64 |
| **Total** | **Full SBFA Execution** | **707.85** | **1068.93** |

Table 4 presents a breakdown of SBFA's runtime across four phases for the Qwen3-1.7B and Qwen3-14B models, evaluated on an NVIDIA A100 (80GB) GPU. The phases include: (1) data/model loading and initial setup, (2) gradient calculation, (3) ranking top-$k$ candidates using SKIP Search, and (4) evaluation of candidate flips. The result shows under single bit flip. If additional flips are necessary, phases 2–4 are repeated in each subsequent iteration. Overall, SBFA exhibits reasonable scalability with model size, requiring approximately 708 seconds for Qwen3-1.7B and 1069 seconds for Qwen3-14B. Notably, the SKIP Search phase (Phase 3) remains efficient even for the larger model, highlighting the effectiveness of the selective skipping strategy in reducing computational overhead. The evaluation phase (Phase 4) is the most time-consuming, as it involves evaluating 100 examples across 100 candidate flips, which is essential for accurate impact estimation. Despite the model size increasing by over $8\times$ from 1.7B to 14B parameters, the total runtime increases by only $1.5\times$, demonstrating that SBFA scales efficiently to larger LLMs.

## 5.5 DISTRIBUTION OF CRITICAL BIT FLIPS

The distribution of critical parameters identified by SBFA is visualized in Figure 3 for LLaMA3.1-8B-Instruct and Gemma3-12B. The x-axis represents the layer index, while the y-axis denotes the number of critical bits found in each layer. The color of each bar indicates the corresponding parameter type (e.g., `mlp.down_proj`, `post_attention_norm`, etc.). As shown in the figure, critical bits are not uniformly distributed across layers or parameter types. Instead, they cluster in specific layers and components, suggesting that certain parts of the model are inherently more vulnerable to SBFA. For example, in LLaMA3.1-8B-Instruct, a substantial number of critical bits are concentrated in layer 1, particularly in the `mlp.down_proj` weights. Likewise, Gemma3-12B exhibits clusters of critical bits in layers 11–17. To further investigate the distribution of vulnerability, we conduct an experiment excluding the `layer.1.mlp.down_proj` component from the attack search in LLaMA3.1-8B-Instruct; the results are shown in Fig. 4 in Appendix B. Remarkably, even after ignoring all weights from this component, SBFA still identifies single-bit flips in other layers or components that reduce accuracy to near zero. Specifically, we find 18 critical flips, predominantly

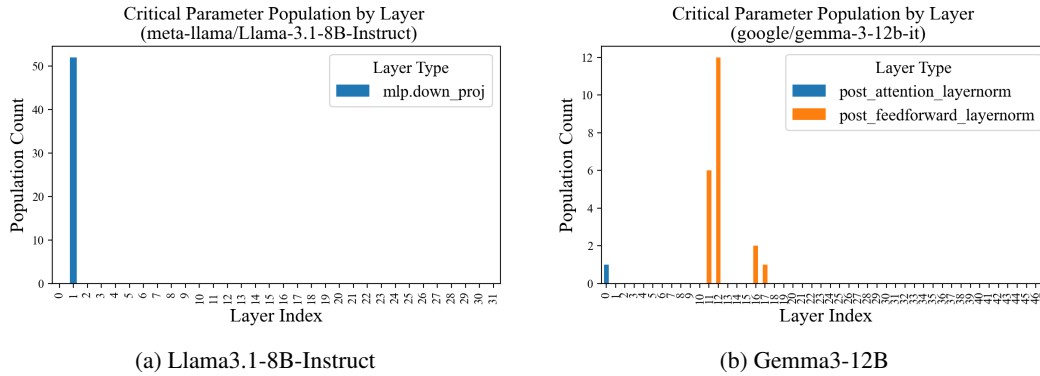

(a) Llama3.1-8B-Instruct

(b) Gemma3-12B

Figure 3: Distribution of critical parameters: (a) Llama3.1-8B-Instruct and (b) Gemma3-12B.

in `layer.0.mlp.down_proj`. This demonstrates that model vulnerability is not confined to a single layer or component, indicating that protecting one part alone may not substantially improve robustness. Further research is needed to enhance the resilience of LLMs against such attacks.

### 5.6 TRANSFER ATTACK

Table 5: Transferability results of models targeted on MMLU to other datasets.

| Model | MMLU → SST-2 | | MMLU → GSM8k | | MMLU → ARC-Easy | |
|---|---|---|---|---|---|---|
| | Pre-ACC | Post-ACC | Pre-ACC | Post-ACC | Pre-ACC | Post-ACC |
| Llama3.1-8B-Instruct | 0.89 | 0.00 | 0.23 | 0.00 | 0.86 | 0.01 |
| Qwen3-8B | 0.95 | 0.00 | 0.62 | 0.00 | 0.92 | 0.00 |

We also evaluate the transferability of SBFA across tasks, with accuracy measured by exact match of generated outputs. Specifically, we test whether bit flips identified on the MMLU dataset can also degrade performance on other datasets, including SST-2, GSM8K, and ARC-Easy. Table 5 summarizes the results for LLaMA3.1-8B-Instruct and Qwen3-8B under BF16 precision. For both models, bit flips from MMLU consistently transfer to other tasks, often driving accuracy to near-zero or causing substantial drops. This demonstrates that the critical bits found by SBFA have a strong transferability across a wide range of tasks, including sentiment analysis (SST-2), common-sense reasoning (ARC-Easy), and mathematical problem solving (GSM8K). It highlights the broad vulnerability of LLMs to targeted bit-flip perturbations.

## 6 CONCLUSION

This paper presents SBFA, a novel and efficient method for conducting bit-flip attacks on large language models. By integrating a ImpactScore with SKIP Search, SBFA effectively identifies critical bits whose perturbation can drastically degrade model performance. Extensive experiments across multiple LLMs and precision settings demonstrate that SBFA can reduce accuracy to near-zero levels with just a single bit flip, highlighting the extreme vulnerability of modern LLMs to bit-level perturbations. Furthermore, SBFA uncovers a broad surface of critical bits distributed across various layers and parameter types, revealing that model susceptibility is not confined to specific components. The identified bit flips also exhibit strong transferability across different tasks, underscoring the generality of the vulnerabilities exposed. Overall, SBFA provides a powerful tool for evaluating and understanding the robustness of large language models against bit-level attacks, with important implications for developing more secure and resilient AI systems.

AI ASSISTANCE DISCLOSURE

We used ChatGPT for grammar and language polishing only (OpenAI, 2025). All technical content is the authors' own.

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

## A  SKIP Search Algorithm

---

**Algorithm 1:** SKIP Search: Selective sKipping for Impact Prioritization

---

**Input:** Model with $L$ layers; top-$k$ size $K$
**Output:** Priority queue $\mathcal{Q}$ with top-$k$ bit flips by ImpactScore
Compute weight ranges $|w_{max}^l - w_{min}^l|$ for each layer $l \in \{1, \dots, L\}$;
Obtain gradients $\nabla w_i^l$ for all weights $w_i^l$;
Initialize empty priority queue $\mathcal{Q}$;
**for** $l \leftarrow 1$ **to** $L$ **do**
    Sort weights $\{w_i^l\}$ in descending order of $|\nabla_{w_i^l}\mathcal{L}|$;
    **for** each weight $w_i^l$ in sorted order **do**
        **if** $|\nabla_{w_i^l}\mathcal{L}| \cdot (\Delta w_{\max}^l - \Delta w_{\min}^l) < \mathcal{Q}_k^{score}$ *and* $|\mathcal{Q}| > K$ **then**
            **break**                // skip remaining weights in this layer
        Compute ImpactScore $= \max_b \left( |\nabla_{w_i^l}\mathcal{L}| \cdot |\Delta w_{i,b}^l| \right)$;
        **if** *ImpactScore* $> \mathcal{Q}_k^{score}$ *or* $|\mathcal{Q}| < K$ **then**
            Insert ImpactScore into $\mathcal{Q}$;
            Prune $\mathcal{Q}$ to retain only top-$K$;

**return** $\mathcal{Q}$;

---

For completeness, we provide the full pseudocode of SKIP Search, reproduced here as Algorithm 1, which was introduced in Section 4.3 of the main text.

## B  Additional Distribution of Critical Bit-Flips

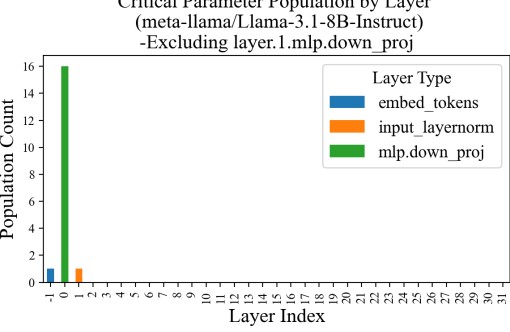

Figure 4: Distribution of critical parameters for Llama3.1-8B-Instruct excluding layer.1.mlp.down proj.

This supplementary experiment demonstrates that even after removing the dominant vulnerable component, SBFA still discovers critical single-bit flips in other layers and components, confirming that vulnerabilities are not confined to a single part of the model.

## C  Extended Experiments on Comparison with Prior Works

To further verify the stability and variance of our method, we include an extended version of Tables 2 and 3 results from all three random seeds. As clearly illustrated, SBFA consistently achieves successful single-bit flips across all runs, with only minor variations in the number of critical single-bit flips. For example, in the LLaMA3-8B-Instruct INT8 setting, all three runs successfully identified single-bit flips, while the number of critical-1-flip cases varied slightly—from $39 \rightarrow 33$ on MMLU and $3 \rightarrow 2$ on SST-2. These results confirm that, even without selecting the best-performing run, SBFA remains stable and consistently superior to all baseline methods.

| Model | Method | MMLU | | | SST-2 | | |
|---|---|---|---|---|---|---|---|
| | | Post-ACC | #Flip | #Crit-1Flip | Post-ACC | #Flip | #Crit-1Flip |
| Llama3-8B-Instruct | GenBFA[†] | 0.00 | 3 | F | - | - | - |
| | BFA (No-Range)[*] | 0.02/0.00/0.00 | 5/3/3 | F/F/F | 0.94/0.88/0.91 | 2/2/3 | F/F/F |
| | BFA (In-Range)[*] | 0.14/0.07/0.03 | 29/7/36 | F/F/F | 0.00/0.00/0.00 | 37/37/31 | F/F/F |
| | **SBFA (Ours)** | 0.00/0.00/0.00 | 1/1/1 | 39/39/33 | 0.00/0.00/0.00 | 1/1/1 | 2/2/3 |
| Phi-3-mini-128k-Instruct | GenBFA[†] | 0.00 | 4 | 1 | - | - | - |
| | BFA (No-Range)[*] | 0.23/0.23/0.58 | 2/2/1 | F/F/F | 0/0/0 | 2/2/2 | F/F/F |
| | BFA (In-Range)[*] | 0.23/0.23/0.24 | 49/2/42 | F/F/F | 0/0/0.47 | 1/1/50 | 1/1/F |
| | **SBFA (Ours)** | 0.19/0/0.22 | 1/1/1 | 8/7/3 | 0/0/0 | 1/1/1 | 6/7/5 |

Table 6: Extended comparison of SBFA with prior methods on MMLU and SST-2 for Llama3.1-8B-Instruct and Phi-3-mini-128k-Instruct in INT8. [†]GenBFA results are reported from (Das et al., 2024). [*]Basic gradient-based BFA results are based on our implementation of the method from (Rakin et al., 2019), with and without range-aware constraints.

| Model | Method | SST2 | | |
|---|---|---|---|---|
| | | Post-ACC | #Flip | #Crit-1Flip |
| Qwen-1.8B | BFA (Sign-only)[‡] | 0.94 | 500 | F |
| | BFA (No-Range)[*] | 0.73/0.73/0.79 | 4/4/4 | F/F/F |
| | BFA (In-Range)[*] | 0.01/0.03/0.01 | 12/12/13 | F/F/F |
| | **SBFA (Ours)** | 0.00/0.00/0.06 | 1/1/1 | 72/74/71 |

Table 7: Extended comparison of SBFA with prior methods on SST2 for Qwen-1.8B in FP32. [‡]BFA results are reported from (Almalky et al., 2025). [*]Basic gradient-based BFA results are based on our implementation of the method from (Rakin et al., 2019), with and without range-aware constraints.

