# OpenReview forum: "SBFA: Single Sneaky Bit Flip Attack to Break Large Language Models"
_ICLR.cc/2026/Conference — Submitted to ICLR 2026_

### Official Review · Reviewer_zDgo · 2025-10-26

**Soundness:** 4
**Presentation:** 4
**Contribution:** 2
**Rating:** 6
**Confidence:** 3

**Summary:**

They propose bit-flip attack aganst LLM, that completely collapses performance with only one bit flip. Unlike a trivial attack that makes a value as extreme as leading to invalid tensor values, they propose a sneaky attack, where the perturbed parameter still remains within a benign distribution.

**Strengths:**

1. **Good presentation**: Their motivation is clear, comparison with related works is reasonably extensive, and each component of their method is well justified (e.g., SKIP Search can reduce the search range from 14B to 15k (line 245)).

2. **Thorough analysis**: Their model choice allows them to analyze their method deeply (e.g., having mixed observations depending on the model family (line 330), and a good scalability to larger models (Table 4)). Further, the analysis is not only extensive, but also “to-the-point”, and readers would be interested in every experimental result that they present.

**Weaknesses:**

1. **Literature review on existing BFAs on LLMs** (line 120-127): The authors claim existing methods are constrained to specific formats, while their method handles diverse formats. I don’t think this is the best framing. For example, given that the method of *Almalky et al., 2025* is to flip the sign bit (line 115), I do not see why this approach does not generalize to int8, which also has a sign bit.

2. **Defense missing**: I think the attack experiments are already quite extensive. Here, I would appreciate it more if there were at least one paragraph that discusses potential defense. For example: (i) Are there any methods that could make the LLM robust against such attacks? (ii) Can the model provider identify potential vulnerable parameters a priori, i.e., are there any properties that are shared across parameters/bits detected through SBFA?

**Questions:**

**Justification on sneakiness**: They claim that their attack being “sneaky (as e.g., depicted in Fig. 2)” is one of their contribution. They show that even under this constraint, the method achieves near-random generation. Here, I am not convinced how stealthier it is compared to extreme perturbations that could yield NaN/Inf (i.e., the ones that the authors mention in their abstract, criticizing that they are not stealthy). For example, I can consider it to be stealthy if the generation is reasonably plausible (e.g., free-form generation that seems plausible but contains factual errors). In contrast, their results demonstrate 0% accuracy on multiple-choice benchmarks. While I understand that ensuring it being within sneaky range is “technically” more challenging and interesting, I would like to ask the authors to elaborate on a “practical” implication of pursuing 0% accuracy instead of ending up with having NaN/Inf.

---

> ### Author Response · Authors · 2025-11-21
> **Response to Reviewer zDgo: Literature, Defense, and Justification on “Sneakiness”**
>
> We thank the reviewer for their insightful comments and for recognizing the clarity and depth of our analysis. Below, we address each point in detail.
>
> **1. Clarification on Almalky et al. (2025) and Format Generalization**
>
> The method proposed by **Almalky et al. (2025)** focuses solely on flipping the MSB(most significant bit). Their experiments in floating-point formats suggested that LLMs exhibit relative tolerance to such attack, with no obvious performance degradation. Almalky et al. did not report results on INT8 models. Our SBFA, in contrast, reports both floating-point and quantized formats seamlessly under the same framework.
>
> **2. Discussion on Defense Methods**
>
> Our current work focuses primarily on demonstrating the **attack feasibility and methodology**. Exploring defense mechanisms will be explored in our future research.
>
> **3. Justification on “Sneakiness”**
>
> The primary purpose of the **sneaky range constraint** (Eq. 1, Fig. 2) is to ensure that flipped weights remain within the **benign layer-wise weight distribution**, thereby preventing extreme numerical errors such as **NaN** or **Inf**, which could crash the system or immediately expose the attack. Here, *“sneaky”* refers specifically to the **bit flip itself**—it ensures that the perturbation remains numerically valid. The **objective of our attack is not a stealthy attack**; rather, the *sneaky bit flip* design simply guarantees that the attack does **not trigger numerical anomalies** such as NaN/Inf values, runtime crashes, or abnormal tensor ranges.
>
> The **practical implication** of pursuing **0% accuracy** rather than triggering NaN/Inf errors is that SBFA causes a **silent functional failure** instead of a **detectable runtime failure**. The compromised model operates without raising error logs or alarms. This property makes SBFA particularly dangerous in real-world deployments, where runtime stability is often monitored but semantic correctness is not. More importantly, our proposed attack method reveals the important vulnerability study of LLM against single bit adversarial weight fault, where such bit-flip caused fault is within the sneaky range constraint as we defined in Eq.1.
>
> We appreciate the reviewer’s comment and our future research will explore **semantic stealthiness**, where the model continues to generate plausible but factually incorrect outputs.

---

> > ### Comment · Reviewer_zDgo · 2025-11-27
> > **Response to authors**
> >
> > I thank the authors for addressing my comments. Here is my response to each of the points.
> >
> > 1. When I first reviewed the paper, it said "(Almalky et al is) restricted the attack to flip the sign bit in floating-point weights", but now it says "restricted the attack to the most significant bit (MSB) in floating-point weights" (line 115). In general, I think the authors updated the PDF without notifying/highlighting the change to reviewers at least twice, as far as I noticed. This makes a proper review significantly difficult.
> >
> > 2. I think currently the attack is, in a way, "too successful," and I think one defense (or something in line to understand why/when the attack works more/less, such as the ones I listed in my review) would add depth to the work. However, I agree with the authors in that this is an attack paper. Therefore, I do not consider this to be a major weakness.
> >
> > 3. Now I understand there is some practical difference between NaN/Inf and 0% accuracy. I appreciate for clarification.

---

> > > ### Author Response · Authors · 2025-11-27
> > > **Response to Reviewer zDgo: Clarifications on MSB**
> > >
> > > We thank the reviewer for their response.
> > >
> > > Regarding point 1, we believe that **“MSB”** is a more appropriate term, as the most significant bit (MSB) in **floating-point** representation corresponds to the **sign bit**. Both terms were used interchangeably in Almalky et al.’s paper

---

### Official Review · Reviewer_QwQA · 2025-10-27

**Soundness:** 3
**Presentation:** 3
**Contribution:** 3
**Rating:** 6
**Confidence:** 3

**Summary:**

This paper propose sneaky bit flip attack (SBFA) for LLM, which can collapses LLM performance with only one single bit flip. SBFA propose ImpactScore to evaluate gradient sensitivity while considering the benign layer-wise weight distribution for stealthiness. The search strategy SKIP also proposed to improve the searching efficiency. SBFA is effective for both floating-point and integer quantized models.

**Strengths:**

- The writing and presentation of the paper are excellent, with a clear and logical flow.
- The proposed attack is stealthy considering the layer-wise weight distribution and effective for both floating-point and integer quantized models.
- The proposed search strategy is effective, demonstrating the practical of SBFA.

**Weaknesses:**

- Only easy tasks have been evaluated. BFA was initially proposed for classification tasks with simple DNN models, such as CNN. However, the major task for LLMs is not classification, but generation task, how about the generation performance influenced by bit flipped.
- No defense methods have been considered.
- No quantitative ablation study to prove the effectiveness of the proposed ImpactScore and SKIP algorithm. The author should further include an ablation study to demonstrate the effect of the benign range (e.g., comparing it with only using the gradient). Table4 only provides the time for each process, but not give the baseline time without using SKIP.

**Questions:**

- In table1, for the mixed setting, there is a lack of vulnerability analysis across different tensor types. Do all Crit-1 Flip bits come from Int8 tensor or floating-point tensor? Compared to Int8 setting, are the int8 tensors in the mixed setting more vulnerable or more robust?
- In table1, for Qwen3-1.7B, why the mixed version is more vulnerable than even BF16, with Crit-1 Flip of 62.
- Have you ever compare the base model and instruct-tuned model to show which is more vulnerable for bit flip attack?

---

> ### Author Response · Authors · 2025-11-21
> **Author Response to Reviewer QwQA: Clarifications on Generation Tasks, Ablation Study, and Model Vulnerability Analysis**
>
> We thank the reviewer for their constructive comments and valuable feedback. Below, we address each point in detail.
>
> **1. Evaluation on Generation Tasks**
>
> We appreciate the reviewer’s concern regarding evaluation on generation-based tasks. In Table 5, we already included results on **GSM8K**, a math word problem benchmark that requires multi-step reasoning and **free-form generation**. As shown, our SBFA achieves **0 accuracy** on GSM8K under the same single-bit flip setting, demonstrating that the proposed attack is also highly effective for generation-based reasoning tasks.
>
> **2. Quantitative Ablation on SKIP Algorithm**
>
> Regarding the SKIP algorithm, Section 4.3 already provided a quantitative reference where the number of evaluated weights on **Qwen3-14B** is reduced from **14.8 billion to only 15,569**, achieving a **~10⁶× reduction** in search complexity. Without SKIP, the search in Step 3 of Table 4 would scale the same amount by ~**10⁶** and require **over 4 years** to complete, compared to only **approximately 16 minutes** with SKIP enabled.
>
> **3. Vulnerability Across Tensor Types (INT8 vs FP)**
>
> In the mixed setting, we manually analyzed all identified Crit-1Flip cases across models and seeds. Among **42 total Crit-1Flip** events, **16** occurred in **INT8 tensors**, while **26** were from **floating-point tensors** (BF16/FP16). This suggests that while both tensor types are vulnerable, floating-point tensors have more number of Crit-1Flip, likely due to their broader dynamic range.
>
> **4. Vulnerability Difference in Qwen3-1.7B (Mixed vs BF16)**
>
> The observed more number of **Crit-1Flip** of the **Qwen3-1.7B mixed** setting compared to BF16 arises from **random seed variation** during dataset evaluation. Different seeds slightly alter the dataset partition and gradient patterns, influencing the identified bit positions and resulting in variability across runs.
>
> **5. Base vs. Instruct-Tuned Models**
>
> We conducted experiments comparing **base** and **instruction-tuned** models, specifically evaluating **Qwen3-8B** and **Qwen3-4B** under different precision settings. All models were susceptible to single-bit attacks across the three precision formats, except for **Qwen3-4B (Instruct)** in the INT8 only setting, as already reported in Table 1. Notably, the **base models** showed more Crit-1Flips: for **Qwen3-8B**, the average number of Crit-1Flips was **47.4** compared to **13.88** for the instruct-tuned model; for **Qwen3-4B**, it was **35.3** versus **8**. So both the base and instruction-tuned models are vulnerable.

---

> > ### Comment · Reviewer_QwQA · 2025-11-26
> >
> > Thank you for your detailed response.  Q1, Q2, Q3, and Q5 are clear. For Q4, given the significant variance caused by random seeds, how stability of the proposed method?
> > What's more, the author claims that this work focuses on demonstrating the attack feasibility and methodology, some defense methods should be considered to highlight the feasibility of SBFA. The author should provide some experiments to demonstrate that the attack cannot be easily defended against.

---

> > > ### Author Response · Authors · 2025-11-26
> > > **Author Response to Reviewer QwQA: Clarifications on Variance of Random Seed and Defense Study**
> > >
> > > We thank the reviewer for their constructive comments and valuable feedback.
> > >
> > > 1. To verify the **stability and variance** of our method, we include an extended version of *Tables 2 and 3* from all three random seeds. As clearly illustrated, **SBFA consistently achieves successful single-bit flips** across all runs, with only minor variations in #Crit-1Flip. For example, in the *LLaMA3-8B-Instruct*, all three runs successfully identified single-bit flips, while the #Crit-1Flip varied slightly—from **39 → 33** on MMLU and **3 → 2** on SST-2. These results confirm that, even without selecting the best-performing run, **SBFA remains stable and consistently superior** to all baseline methods.
> > > **Table: Comparison of SBFA with Prior Methods on MMLU and SST-2**
> > >
> > > | **Model** | **Method** | **MMLU Post-ACC** | **#Flip** | **#Crit-1Flip** | **SST-2 Post-ACC** | **#Flip** | **#Crit-1Flip** |
> > > |------------|-------------|-------------------|------------|------------------|---------------------|------------|------------------|
> > > | **Llama3-8B-Instruct** | GenBFA† | 0.00 | 3 | F | 0.94 | 2 | - |
> > > | | BFA (No-Range)* | 0.02 / 0.00 / 0.00 | 5 / 3 / 3 | F / 1 / 1 | 0.94 / 0.88 / 0.91 | 2 / 2 / 3 | F / F / F |
> > > | | BFA (In-Range)* | 0.14 / 0.07 / 0.03 | 29 / 17 / 36 | F / F / F | 0.00 / 0.00 / 0.00 | 37 / 37 / 31 | F / F / F |
> > > | | **SBFA (Ours)** | **0.00 / 0.00 / 0.00** | **1 / 1 / 1** | **39 / 39 / 33** | **0.00 / 0.00 / 0.00** | **1 / 1 / 1** | **2 / 2 / 3** |
> > > | **Phi-3-mini-128k-Instruct** | GenBFA† | 0.00 | 4 | 1 | - | - | - |
> > > | | BFA (No-Range)* | 0.23 / 0.23 / 0.58 | 2 / 2 / 1 | F / F / F | 0 / 0 / 0 | 2 / 2 / 2 | F / F / F |
> > > | | BFA (In-Range)* | 0.23 / 0.20 / 0.24 | 49 / 2 / 42 | F / F / F | 0 / 0 / 0.47 | 1 / 1 / 50 | 1 / 1 / F |
> > > | | **SBFA (Ours)** | **0.19 / 0.00 / 0.22** | **1 / 1 / 1** | **8 / 7 / 3** | **0 / 0 / 0** | **1 / 1 / 1** | **6 / 7 / 5** |
> > >
> > > † GenBFA results are reported from Das et al. (2024).
> > > \* BFA results are based on our reimplementation of Rakin et al. (2019), with and without range constraints.
> > >
> > > **Table: Comparison of SBFA with Prior Methods on SST-2 for Qwen-1.8B**
> > >
> > > | **Model** | **Method** | **SST-2 Post-ACC** | **#Flip** | **#Crit-1Flip** |
> > > |------------|-------------|--------------------|------------|------------------|
> > > | **Qwen-1.8B** | BFA (Sign-only)‡ | 0.94 | 500 | F |
> > > | | BFA (No-Range)* | 0.73 / 0.73 / 0.79 | 4 / 4 / 4 | F / F / F |
> > > | | BFA (In-Range)* | 0.01 / 0.03 / 0.01 | 12 / 12 / 13 | F / F / F |
> > > | | **SBFA (Ours)** | **0.00 / 0.00 / 0.06** | **1 / 1 / 1** | **72 / 74 / 71** |
> > >
> > > ‡ BFA (Sign-only) results are reported from Almalky et al. (2025).
> > > \* BFA (No-Range) and BFA (In-Range) are based on our reimplementation of Rakin et al. (2019).
> > >
> > > 2. For defense study, we already discussed in Section 5.1. In our defense study, we ‘protected’ the sensitive component(layer.1.mlp.down.proj) for Llama3.1-8B-Instruct, assuming no fault could be injected for this component, as it produced the most Crit-1Flip cases. Then, we applied our SBFA on remaining ‘unprotected’ components. It is interesting that SBFA could find different 18 critical flips in the remaining unprotected components, indicating that simply protecting the sensitive component is not enough to defend against SBFA. More research on the defense is required for future work.

---

### Official Review · Reviewer_E1du · 2025-10-29

**Soundness:** 2
**Presentation:** 3
**Contribution:** 3
**Rating:** 4
**Confidence:** 3

**Summary:**

This paper studies bit-flip attacks against LLMs and proposes a new framework called SBFA. The goal is to show that flipping a single bit in the LLM's weights can catastrophically degrade model performance. The authors observe that previous BFAs often cause numerical overflow (NaN) in floating-point formats, making the attack easy to detect. To address this, SBFA introduces a "stealthy flip" constraint that ensures the flipped weight remains within the layer’s normal range. The paper defines an ImpactScore, calculated as the product of the gradient magnitude and the maximum feasible perturbation under this constraint, to rank bit candidates. A SKIP strategy is also proposed to skip unpromising layers based on upper bounds, significantly speeding up the search. Experiments show that SBFA can drive model accuracy to near zero across multiple models (Qwen, LLaMA, Gemma) and formats (BF16, INT8), with claimed cross-task transferability from MMLU to SST-2, GSM8K, and ARC-Easy.

**Strengths:**

- The topic itself, i.e., studying the security of large language models, is important and timely.
- The paper’s observations and methods are interesting. The observation that previous work suffered from numerical overflow issues is interesting, and the proposed constraint that enforces flipped weights to stay within the normal range is reasonable. The proposed ImpactScore and SKIP strategy are also intuitively sound and smartly designed.
- The experimental results are quite strong. The paper claims flipping one-bit can work across multiple models and tasks and shows a large number of "Crit-1Flip" cases and cross-task transferability. Such results would be a serious warning to the community.

**Weaknesses:**

- The threat model is quite strong and somewhat unrealistic. The attack assumes full white-box access, the ability to compute gradients, and precise control to flip a specific bit in the deployed environment. These assumptions are very strong, and the paper does not describe the realistic scenarios where they hold. Since the attack completely destroys the model’s usability, even simple validation checks could easily detect and replace the corrupted model. It is unclear what the attacker’s motivation would be and why the victim would continue using such a corrupted model.
- The paper assumes that an attacker can flip a specific bit corresponding to a model weight through Rowhammer, but modern DRAM (e.g., DDR5) includes strong mitigations, which make this extremely hard. Recent studies [1,2] show that modern Rowhammer attacks require extensive reverse engineering of hardware address mappings and precise timing synchronization, which might be orders of magnitude more complex than what this paper assumed.
- In Section 5.1, the authors write "All results are reported from the best outcome over three runs with different predefined random seeds", meaning they report only the best of three runs. It is unclear why this choice is made, as it may overestimates the attack strength and also raises concerns about fairness when compared to baselines. If the proposed method exhibits high variance, while baselines are more stable, this evaluation protocol could make SBFA look stronger simply due to lucky runs rather than consistent superiority.

Ref:

[1] Kim et al. Rowhammer attacks in dynamic random-access memory and defense methods

[2] Jattke et al. McSee: Evaluating Advanced Rowhammer Attacks and Defenses via Automated DRAM Traffic Analysis

**Questions:**

NA

---

> ### Author Response · Authors · 2025-11-22
> **Author Response to Reviewer E1du: Clarifications on Threat Model Realism and Evaluation Protocol**
>
> We sincerely thank the reviewer for their thoughtful feedback and helpful suggestions. Below, we provide clarifications in detail.
> **1. Realism of the Threat Model**
>
> Our work follows the **standard threat model in prior BFA literatures** [1,2,3] and are widely used to analyze *vulnerability and robustness* of neural networks. Our goal is not to propose a new Rowhammer attack mechanism, but rather to investigate **how catastrophic even a single bit flip could be** if such a fault were to occur in a deployed LLM.
>
> Importantly, recent studies have demonstrated that **Rowhammer-based bit flips remain practical even on modern DDR4/5 hardware**. For example, **DeepHammer** [3] achieved deterministic bit flips on real DNN systems under DDR3/4. The recent **REFault** study [4] demonstrates that DDR5 DRAM cells remain physically susceptible to Rowhammer, showing that the observed resistance of DDR5 stems from improved in-DRAM mitigations rather than a more resilient memory substrate—implying that Rowhammer attacks could still be possible if these defenses are bypassed. On top of that, **Phoenix** [5], which was from the author group of **McSee[6]** the reviewer cited, demonstrated the *first practical Rowhammer attack on DDR5*, reliably inducing bit flips on all tested DIMMs and escalating privileges to root within approximately 109 seconds. These findings confirm that, despite in-DRAM mitigations, **DDR5 remains vulnerable** to unprivileged Rowhammer attacks. Therefore, our assumed Rowhammer-based fault model is **technically feasible and security-relevant**.
>
> **2. Lack of practical motivation**
>
> Our attack objective is to malfunction the LLM with one critical bit flip. Our motivation aligns with prior BFA studies, which aim to expose and quantify the vulnerability of AI models to hardware-level integrity faults. A practical example would be an attacker targeting a service provider to deliberately corrupt a deployed model at a chosen time. Such an attack could disrupt user access, degrade service quality, and significantly damage the provider’s reliability and reputation. Thus, understanding these vulnerabilities is critical.
>
> **3. Justification for Reporting “Best of Three Runs”**
>
> In adversarial attack research, evaluating with **multiple runs/restarts** and reporting the **strongest attack** is standard practice (e.g., Siddiqui et al. explicitly pick the best adversarial example from 100 restarts; DeepHammer reports the minimum number of bit-flips)**[3,7]**, as the goal is to evaluate the **maximum capability** of a determined adversary.
>
> To further verify the **stability and variance** of our method, we include an extended version of *Tables 2 and 3* from all three random seeds in the follow-up response. As clearly illustrated, **SBFA consistently achieves successful single-bit flips** across all runs, with only minor variations in #Crit-1Flip. For example, in the *LLaMA3-8B-Instruct*, all three runs successfully identified single-bit flips, while the #Crit-1Flip varied slightly—from **39 → 33** on MMLU and **3 → 2** on SST-2. These results confirm that, even without selecting the best-performing run, **SBFA remains stable and consistently superior** to all baseline methods.
>
> **4. Our Contribution and Impact**
>
> In summary, our findings directly reinforce the **core contribution and significance** of this work. Our work presents the **first demonstration** of a single bit flip **completely compromising state-of-the-art LLMs** such as LLaMA3, Qwen3, and Gemma across both BF16 and INT8 formats. The proposed **ImpactScore** metric and **SKIP Search** algorithm together achieve high **attack efficiency**, identifying critical bits within **minutes** on billion-parameter models.
>
> Beyond methodological contributions, our findings serve as a **critical warning** that even a single adversarial weight fault can lead to catastrophic, system-wide degradation in deployed LLMs—emphasizing the urgent need for **fault-resilient model design and memory protection mechanisms** in future AI infrastructure.
>
> [1] Rakin, et al. "Bit-flip attack: Crushing neural network with progressive bit search." *ICCV*. 2019.
>
> [2] Rakin et al. "Tbt: Targeted neural network attack with bit trojan." ICCV, 2020.
>
> [3] Yao, et al. "{DeepHammer}: Depleting the intelligence of deep neural networks through targeted chain of bit flips." *USENIX,* 2020.
>
> [4] Gloor et al. "REFault: A Fault Injection Platform for Rowhammer Research on DDR5 Memory." uASC, 2025.
>
> [5] Meyer, Diego, et al. "Phoenix: Rowhammer Attacks on DDR5 with Self-Correcting Synchronization." *S&P* (2026).
>
> [6] Jattke, Patrick, et al. "{McSee}: Evaluating Advanced Rowhammer Attacks and Defenses via Automated {DRAM} Traffic Analysis." *USENIX,* 2025.
>
> [7] Siddiqui et al. "Benchmarking adversarial attacks and defenses for time-series data." *International Conference on Neural Information Processing*. Cham: Springer International Publishing, 2020.

---

> > ### Author Response · Authors · 2025-11-22
> > **Additional Material: Extended Comparison Tables**
> >
> > We added the two extended comparison tables below in this follow-up comment due to character limitations in the main response. Those tables are also in the appendix.
> > **Table: Comparison of SBFA with Prior Methods on MMLU and SST-2**
> >
> > | **Model** | **Method** | **MMLU Post-ACC** | **#Flip** | **#Crit-1Flip** | **SST-2 Post-ACC** | **#Flip** | **#Crit-1Flip** |
> > |------------|-------------|-------------------|------------|------------------|---------------------|------------|------------------|
> > | **Llama3-8B-Instruct** | GenBFA† | 0.00 | 3 | F | 0.94 | 2 | - |
> > | | BFA (No-Range)* | 0.02 / 0.00 / 0.00 | 5 / 3 / 3 | F / 1 / 1 | 0.94 / 0.88 / 0.91 | 2 / 2 / 3 | F / F / F |
> > | | BFA (In-Range)* | 0.14 / 0.07 / 0.03 | 29 / 17 / 36 | F / F / F | 0.00 / 0.00 / 0.00 | 37 / 37 / 31 | F / F / F |
> > | | **SBFA (Ours)** | **0.00 / 0.00 / 0.00** | **1 / 1 / 1** | **39 / 39 / 33** | **0.00 / 0.00 / 0.00** | **1 / 1 / 1** | **2 / 2 / 3** |
> > | **Phi-3-mini-128k-Instruct** | GenBFA† | 0.00 | 4 | 1 | - | - | - |
> > | | BFA (No-Range)* | 0.23 / 0.23 / 0.58 | 2 / 2 / 1 | F / F / F | 0 / 0 / 0 | 2 / 2 / 2 | F / F / F |
> > | | BFA (In-Range)* | 0.23 / 0.20 / 0.24 | 49 / 2 / 42 | F / F / F | 0 / 0 / 0.47 | 1 / 1 / 50 | 1 / 1 / F |
> > | | **SBFA (Ours)** | **0.19 / 0.00 / 0.22** | **1 / 1 / 1** | **8 / 7 / 3** | **0 / 0 / 0** | **1 / 1 / 1** | **6 / 7 / 5** |
> >
> > † GenBFA results are reported from Das et al. (2024).
> > \* BFA results are based on our reimplementation of Rakin et al. (2019), with and without range constraints.
> >
> > **Table: Comparison of SBFA with Prior Methods on SST-2 for Qwen-1.8B**
> >
> > | **Model** | **Method** | **SST-2 Post-ACC** | **#Flip** | **#Crit-1Flip** |
> > |------------|-------------|--------------------|------------|------------------|
> > | **Qwen-1.8B** | BFA (Sign-only)‡ | 0.94 | 500 | F |
> > | | BFA (No-Range)* | 0.73 / 0.73 / 0.79 | 4 / 4 / 4 | F / F / F |
> > | | BFA (In-Range)* | 0.01 / 0.03 / 0.01 | 12 / 12 / 13 | F / F / F |
> > | | **SBFA (Ours)** | **0.00 / 0.00 / 0.06** | **1 / 1 / 1** | **72 / 74 / 71** |
> >
> > ‡ BFA (Sign-only) results are reported from Almalky et al. (2025).
> > \* BFA (No-Range) and BFA (In-Range) are based on our reimplementation of Rakin et al. (2019).

---

> > > ### Comment · Reviewer_E1du · 2025-11-28
> > > **Thank you for your rebuttal!**
> > >
> > > Dear authors, thank you very much for the detailed rebuttal! I genuinely appreciate the excellent clarifications and learned quite a bit from your response. Also thanks for pointing me to the Phoenix work, and I will definitely look into it. Your explanation of the attack scenario is also very reasonable and addressed my earlier doubts. I also appreciate the extended experimental results which resolve my questions regarding stability and variance. Nice job here.
> > >
> > > Given that your rebuttal has addressed nearly all of my concerns very clearly, I plan to raise my score to an 8. It seems the system does not allow editing at the moment, but I will update the score once it becomes available. Thanks!

---

### Official Review · Reviewer_b82p · 2025-10-31

**Soundness:** 3
**Presentation:** 3
**Contribution:** 3
**Rating:** 6
**Confidence:** 3

**Summary:**

This paper introduces SBFA (Sneaky Bit-Flip Attack), a framework that investigates the vulnerability of large language models (LLMs) to single-bit flips. The authors first propose ImpactScore, a metric designed to assess the sensitivity and vulnerability of individual parameters. Based on this metric, they develop an efficient search strategy, SKIP (Selective sKipping with Impact Prioritization) Search, to identify impactful bit-flips. Extensive experiments demonstrate the effectiveness of SBFA on both BF16 and INT8 formats across a variety of LLMs and benchmarks.

**Strengths:**

1. The paper is well written and easy to follow. The SBFA framework, along with the ImpactScore metric and SKIP search algorithm, is described clearly and in detail.

2. The experimental evaluation is comprehensive. The authors test SBFA on a broad range of widely used LLMs, including Qwen2.5-7B, Qwen3 series, LLaMA3-18B-Instruct, and Gemma3-12B. The results convincingly demonstrate the effectiveness of the proposed approach.

**Weaknesses:**

1. The runtime overhead of SBFA is significant. Identifying candidate bit flips is computationally expensive, taking more than half of the total execution time as shown in Table 4.

2. The paper lacks a direct comparison with prior work in terms of runtime efficiency and transferability, which would help position SBFA more clearly in the existing literature.

**Questions:**

1. How does task difficulty affect the transferability of bit-flip attacks? For instance, Table 5 shows that critical bits identified using the multi-task MMLU benchmark can transfer effectively to both simple (SST-2) and hard (GSM8K) tasks, but not to medium-difficulty tasks such as commonsense reasoning. Can the authors provide insights into this behavior?

2. How does SBFA compare to previous approaches in terms of both efficiency and transferability? A quantitative comparison would strengthen the experimental section.

---

> ### Author Response · Authors · 2025-11-21
> **Author Response to Reviewer b82p: Clarifications on Transferability, Efficiency, and Runtime Analysis**
>
> We sincerely thank the reviewer for their thoughtful feedback and helpful suggestions. Below, we provide clarifications and additional analysis to address the raised concerns.
>
> **1. Transferability Across Task Difficulty**
>
> Regarding the reviewer’s question about task difficulty and transfer behavior, we appreciate the insight. As shown in *Table 5*, SBFA demonstrates strong transferability from **MMLU** to both **SST-2** (simple classification) and **GSM8K** (complex generation-based reasoning). The relatively weaker transfer to **ARC_Easy** arises from differences in the **evaluation metric** used by the **LM Harness** library: ARC_Easy is evaluated based on **likelihood accuracy**, whereas other datasets use **exact match from generated output**. When we switch ARC_Easy’s evaluation to **generated exact match**, both **Llama3.1-8B-Instruct** and **Qwen3-8B** reach to **0 accuracy** under the same bit-flip transferability settings. This confirms that the attack transfers effectively across reasoning, classification, and generation tasks when evaluated under a consistent metric.
>
> Overall, these results demonstrate that SBFA not only scales efficiently but also generalizes effectively across diverse task types and difficulty levels, highlighting its practical relevance and robustness.
>
> **2. Transferability across different methods**
>
> We also conducted transferability tests using the **gradient-only BFA** method on **llama3.1-8B-instruct** and observed **similar results**, namely, the attacks transferred effectively across datasets under the exact match metric. This behavior is consistent with the findings reported in the **GenBFA** paper, where other datasets also achieved **0 accuracy** after transfer. Such outcomes are a **common characteristic of BFA attacks on LLMs**, as high-impact bit flips often lead to widespread degradation across multiple tasks.
>
> **Table: Transferability Results Across Tasks using Exact Match**
>
> | **Method** | **SST-2** | **GSM8K** | **ARC-Easy** |
> |-------------|-----------|-----------|---------------|
> | **SBFA**    | 0.00      | 0.00      | 0.01          |
> | **BFA**     | 0.00      | 0.23      | 0.00          |
> | **GenBFA**  | 0.00      | 0.00      | 0.00          |
> **3. Runtime and Efficiency Comparison**
>
> We evaluate **efficiency** with respect to the following two metrics.:
>
> (1) **The ability to identify critical 1-bit flips**, and
>
> (2) **The overall runtime**.
>
> As shown in *Table 2* from the manuscript, only our SBFA consistently identifies **critical single-bit flips** across all models and datasets. Prior methods such as **GenBFA** fail to achieve single-bit degradation, while **gradient-only BFAs** succeed only occasionally (e.g., Phi-3-mini-128k-Instruct on SST-2). Therefore, SBFA demonstrates substantially higher efficiency in terms of **attack potency**, with only moderate computational overhead compared to gradient-based BFAs.
>
> Regarding **runtime**, SBFA introduces minor additional computational steps compared to traditional gradient-based BFAs. From the table shown below, for the **Qwen3-1.7B** model, gradient-based BFA finishes in **165 seconds**, while SBFA completes in approximately **707 seconds (~12 minutes)**, which is significantly shorter than the **GenBFA** runtime of **18,593 seconds (~5.2 hours)** under the same setting. Thus, although SBFA requires additional processing to ensure bit-flip range and effectiveness for 1-bit flip, its total runtime is not excessive relative to prior approaches and **significantly faster than GenBFA,** confirming that our method achieves **superior overall efficiency** in both **attack effectiveness** and **execution time**.
>
> **Table: Runtime Comparison of Different Attack Methods**
>
> | **Model**     | **GenBFA (s)** | **BFA (s)** | **SBFA (s)** |
> |----------------|----------------|--------------|---------------|
> | Llama3.1 8B    | 23028          | 792          | 1175          |
> | Qwen3 1.7      | 18593          | 165          | 707           |

---

### Meta-Review · Area_Chair_g2Sz · 2026-01-02

**Summary:**

This paper proposes SBFA (Sneaky Bit-Flip Attack), a framework showing that flipping a single carefully selected bit in large language model weights can catastrophically degrade performance while keeping the perturbed value within a benign layer-wise range. Reviewers acknowledge that the topic is important, the technical design (ImpactScore and SKIP search) is well motivated, and the empirical results demonstrate a striking vulnerability of modern LLMs. Several reviewers also found the rebuttal thorough and technically convincing, with clarifications on transferability, runtime efficiency, and threat-model assumptions, and at least one reviewer indicated an intention to significantly raise their score.

However, independent of the technical merits, **this submission violates the double-blind review policy**. As reported by a reviewer and confirmed by the previous Area Chair and escalated to the Senior Area Chairs and Program Chairs, the revised version of the manuscript explicitly included author-identifying information, breaking anonymity. Although the current system-visible version appears anonymized due to rollback or system issues, the violation itself has been confirmed and documented. According to conference policy, such a breach necessitates rejection regardless of technical quality. Therefore, this paper cannot be accepted

**Reviewer Concerns:**

From a technical perspective, reviewers raised and discussed several substantive issues, including the realism of the threat model (white-box access, precise bit control via Rowhammer), the practical attacker motivation, evaluation protocol choices (reporting best-of-several runs), and the absence of defense exploration. The authors’ rebuttal addressed many of these concerns in detail, citing recent DDR5 Rowhammer studies, providing extended stability and transferability results, and clarifying that the goal is to study worst-case vulnerability rather than propose a stealthy end-to-end exploit. Some reviewers indicated that these responses substantially improved their confidence in the work.

Nevertheless, a non-technical but decisive concern supersedes these discussions. A reviewer explicitly reported that the revised PDF contained the authors’ names, and this was confirmed by an Area Chair and escalated to the Senior Area Chairs. This constitutes a direct violation of the double-blind review requirements. Regardless of the subsequent rebuttal quality or reviewer score changes, the presence of author-identifying information in a revised submission compromises the integrity of the review process and renders the paper ineligible for acceptance.

**Reviewer Scores:**

Reviewer scores span from borderline reject to marginally above the acceptance threshold, and the rebuttal led at least one reviewer to explicitly state their intention to raise their score substantially. Under normal circumstances, the paper might therefore be considered a borderline or even weak-accept case based on technical merit alone.

However, due to the confirmed violation of the double-blind review policy, reviewer score dynamics are no longer relevant to the final decision. Even if a full discussion period were held and technical consensus shifted toward acceptance, the anonymity breach would still require rejection. As such, the appropriate and policy-consistent outcome for this submission is rejection, independent of its technical evaluation.

---

### Decision · Program_Chairs · 2026-01-26

Reject